# Stress-Activated Protein Kinase Signalling Regulates Mycoparasitic Hyphal-Hyphal Interactions in *Trichoderma atroviride*

**DOI:** 10.3390/jof7050365

**Published:** 2021-05-06

**Authors:** Dubraska Moreno-Ruiz, Linda Salzmann, Mark D. Fricker, Susanne Zeilinger, Alexander Lichius

**Affiliations:** 1Department of Microbiology, University of Innsbruck, 6020 Innsbruck, Austria; dubraska.moreno-ruiz@uibk.ac.at (D.M.-R.); linda.salzmann@risch.ch (L.S.); Susanne.Zeilinger@uibk.ac.at (S.Z.); 2Department of Plant Sciences, University of Oxford, Oxford OX1 3RB, UK; mark.fricker@plants.ox.ac.uk

**Keywords:** *Trichoderma atroviride*, mycoparasitism, plant pathogens, polarity stress, CRIB reporter, GTPase activity, MAPK/SAPK signalling

## Abstract

*Trichoderma atroviride* is a mycoparasitic fungus used as biological control agent against fungal plant pathogens. The recognition and appropriate morphogenetic responses to prey-derived signals are essential for successful mycoparasitism. We established microcolony confrontation assays using *T. atroviride* strains expressing cell division cycle 42 (Cdc42) and Ras-related C3 botulinum toxin substrate 1 (Rac1) interactive binding (CRIB) reporters to analyse morphogenetic changes and the dynamic displacement of localized GTPase activity during polarized tip growth. Microscopic analyses showed that *Trichoderma* experiences significant polarity stress when approaching its fungal preys. The perception of prey-derived signals is integrated via the guanosine triphosphatase (GTPase) and mitogen-activated protein kinase (MAPK) signalling network, and deletion of the MAP kinases *Trichoderma* MAPK 1 (Tmk1) and Tmk3 affected *T. atroviride* tip polarization, chemotropic growth, and contact-induced morphogenesis so severely that the establishment of mycoparasitism was highly inefficient to impossible. The responses varied depending on the prey species and the interaction stage, reflecting the high selectivity of the signalling process. Our data suggest that Tmk3 affects the polarity-stress adaptation process especially during the pre-contact phase, whereas Tmk1 regulates contact-induced morphogenesis at the early-contact phase. Neither Tmk1 nor Tmk3 loss-of-function could be fully compensated within the GTPase/MAPK signalling network underscoring the crucial importance of a sensitive polarized tip growth apparatus for successful mycoparasitism.

## 1. Introduction

Several species of the filamentous fungal genus *Trichoderma* are facultative plant symbionts in parallel to being highly effective mycoparasites against devastating fungal plant pathogens. In particular, *T. atroviride* is widely used as a general plant growth promoter and biological control agent in modern agriculture [1]. Direct mycoparasitism of phytopathogenic prey fungi is one important feature of biocontrol. The mycoparasitic interaction is characterized by several steps including prey sensing, directional chemotropic approach, pre-contact chemical attack, and direct physical interaction between mycoparasite and prey hyphae. Numerous prey fungi, on the other hand, have the potential to counteract mycoparasitic attack with highly effective defence responses [2,3]. 

The initiating event of prey sensing does not occur in a random manner. It was hypothesised early on and confirmed subsequently that *Trichoderma* species recognize and target their prey by chemotropic growth along gradients of prey-derived compounds [4,5,6]. Hyphal tropism is the result of sensing either positive or negative chemotropic compounds that, respectively, induce attraction (chemoattractants) or repulsion (chemorepellents) [7]. In *T. atroviride*, both hyphal density and colony extension rate are positively affected in the vicinity of chemoattractants, such as the self-signalling compound 6-pentyl-α-pyrone (6-PP) and the plant oxylipin 13S-hydroxy-9Z,11E-octadecadienoic acid (13(s)-HODE) which is released by stressed plants to call in the mycoparasite [8]. 

Initial prey recognition results in transcriptional changes including the increased expression of cell wall degrading (CWD) enzymes [9,10,11], which reinforce the release of compounds from the prey cell wall which then serve as chemoattractants to guide the mycoparasite to its prey. In addition, secreted proteases are thought to release prey-derived oligopeptides, which act as ligands for G protein-coupled receptors (GPCRs) that mediate prey sensing [12]. Such prey-derived diffusible substances trigger the expression of genes that put the mycoparasite into “attack mode” [13,14,15,16].

Once the mycoparasite has located the prey mycelium its hyphal morphology changes in the immediate interaction zone to establish direct physical contact with prey hyphae by multiple branching and coiling, which in some instances may be followed by formation of appressorium-like structures and penetration [17,18,19,20]. 

Tight control over polarised tip growth and branching is essential for filamentous fungi to form and maintain the tubular hyphal shape and to adapt its morphology to intra- and extracellular signals. As in all multicellular eukaryotes, the small Rho-type GTPases Cdc42 and Rac1 are key regulators of hyphal tip polarity [21] and operate by switching between an inactive GDP-bound state and an active GTP-bound state [22,23]. Only active-state GTPases can activate downstream effectors such as p21-activated kinases (PAKs) and mitogen-activated protein kinases (MAPKs) [24] to induce morphogenetic changes via cytoskeletal and exo-/endocytotic rearrangements [25]. The Cdc42/Rac1-interactive binding (CRIB) motif is embedded within the p21-binding domain (PBD) and present in most downstream effectors of Cdc42 and Rac1 [24]. Furthermore, PBD plus associated pleckstrin-homology (PH) or basic-rich (BR) membrane-interaction domains of PAKs [26] are necessary and sufficient for functional association between active GTPase and downstream effectors [27,28]. Utilising this functional relationship, the CRIB reporters that specifically localize to active Rho GTPases [29] were constructed in *Neurospora crassa* and proved instrumental in dissecting shared and distinct roles of CDC42 and RAC1 in the regulation of polarised germ tube growth and conidial anastomosis tube (CAT)-mediated cell fusion, and, also, highlighted the functional connection between GTPase and MAPK signalling in this process [30].

The three classical MAPK pathways regulating pheromone response (PR), cell wall integrity (CWI), and high-osmolarity glycerol (HOG) are represented in *T. atroviride* by the respective terminal MAPKs Tmk1 (cell fusion 3 (Fus3)/kinase suppressor of Sst2 mutations (Kss1) homolog), Tmk2 (suppressor of the LyTic phenotype (Slt2) homolog), and Tmk3 (Hog1 homolog), respectively. Tmk1 is involved in the regulation of conidiation, chitinase and antifungal metabolite production, as well as coiling in *T. atroviride* and *T. virens* [31,32,33]. Tmk2 has been shown in *T. reesei* to regulate cell wall integrity, sporulation, and cellulase production [33,34,35,36]. Tmk3 responds to numerous stress conditions, including osmotic and oxidative stress, heavy metal toxicity, and affects injury abilities in *T. atroviride* and *T. reesei* [37,38,39,40], and is thus a recognised component of the stress-activated protein kinase (SAPK) module [41]. 

With the aim to better understand the underlying GTPase and MAPK/SAPK signalling pathways that may trigger virulence-related morphogenetic changes of individual hyphae, we aimed to address the following key questions: (1) do CRIB reporters label apical GTPase activity clusters in *Trichoderma* during hyphal tip growth including chemotropism, (2) do CRIB reporters facilitate the quantification of morphogenetic changes in individual hyphae of *Trichoderma* during mycoparasitic confrontations, and (3) does the analysis of MAPK gene deletion mutants expressing CRIB reporters reveal an important role for MAPK/SAPK signalling in this process?

## 2. Materials and Methods

### 2.1. Fungal Strains

*Trichoderma atroviride* (strain P1; ATCC 74058), Δ*tmk1* [32], and Δ*tmk3* [42] were used as mycoparasites. The plant pathogens *Botrytis cinerea* B05.10, *B. cinerea* B05.10 dsRed strain [43], *Fusarium oxysporum* f. sp. *lycopersici* strain 4287, and *Rhizopus microsporus* (isolate obtained from Martin Kirchmair, Department of Micobiology, University of Innsbruck) were used as prey fungi. Strains were grown on plain potato dextrose agar (PDA; BD Difco, Franklin Lakes, NJ, USA) or PDA supplemented with 200 µg/mL hygromycin-B (Merck Millipore, Burlington, MA, USA) where appropriate. Growth conditions comprised room temperature (*B. cinerea*) or 25 °C (all other species), and natural or 12/12 h light/dark cycles, respectively.

### 2.2. Dual-Culture Confrontation Assays

Macro- and microcolony confrontation assays between mycoparasite and prey fungi were performed as triplicates on PDA (macrocolony format) and modified M9 (MM9) (microcolony format) agar plates, respectively, as described previously [8]. In short, 5 mm^2^ PDA or MM9 agar plugs containing mycelia were inoculated at a distance of 1.2 cm in the microcolony format or at 6.0 cm in the standard macrocolony format. All cultures were incubated under the respective conditions for a total of 36 h before macroscopic and microscopic analyses.

### 2.3. Cloning of CRIB GTPase-Activity Reporters

Following the same strategy as for *N. crassa* [30], plasmids for high-level cytoplasmic expression of GTPase-activity reporters were generated by first introducing the hygromycin B resistance-mediating cassette (*hph* cassette) from pGFP-XYR1 [44] into pZEGA1 [14] to give pLS1 (P*pki*::*sgfp*) [45]. Secondly, the coding region of the CRIB motif-containing PBD from *T. atroviride’s* Cln activity dependant 4 (Cla4) PAK (ID316062; https://mycocosm.jgi.doe.gov/Triat2/Triat2.home.html accessed on 12 July 2016) was subcloned between the *pki1* promoter and the synthetic green fluorescent protein (sGFP) (S65T) coding region [46] of a linearised pLS1 backbone, yielding pLS1-CRIBc-sGFP (P*pki*::*crib(cla4)-sgfp*). In comparison to the corresponding *N. crassa* strain [30], CRIBc-sGFP reporter fluorescence was noticeably weaker in *T. atroviride*. Therefore, following [47], two site-directed mutagenesis steps were employed to introduce three amino acid exchanges that modified sGFP into a photophysically improved variant which we termed monomeric BasicGFP (mBasicGFP) (Appendix A). The first step involved F64L and S72A exchanges in sGFP using oligonucleotide sGFP-QC-F and its reverse complement (Appendix A), to yield pLS2-CRIBc-BasicGFP (P*pki*::*crib(cla4)-basicgfp*). In the second step, the homodimerisation domain of BasicGFP was disrupted through the A206K exchange using oligonucleotide basicGFP-QC-F and its reverse complement (Appendix A), yielding plasmid pLS3-CRIBc-mBasicGFP (P*pki*::*crib(cla4)-mbasicgfp*). To express the CRIB reporter in the hygromycin B-resistant Δ*tmk1* strain [32], the *hph*-cassette in pLS3-CRIBc-mBasicGFP was exchanged for a *nat1* cassette conferring resistance against 200 µg/mL Nourseothricin (AB-101, Jena Bioscience GmbH, Jena, Germany) [48,49] creating plasmid pDM-*tmk1*-CRIBc. To generate the *tmk3* gene deletion strain of *T. atroviride* expressing the CRIB reporter, plasmid pDM-*tmk3*-CRIBc was constructed to exchange the target locus (ID 301235) with the CRIB reporter expression cassette by homologous recombination. For this, 1 kb-fragments of the 5′ and 3′ noncoding regions of the *tmk3* locus were amplified from genomic DNA using primers 5′Ptmk3-Fw/Rv and 3′Ttmk3-Fw/Rv (Appendix A), respectively, and added to either side of the CRIB reporter expression cassette in pLS3-CRIBc-mBasicGFP. Molecular assembly of the respective inserts and vector backbone fragments into target plasmids was performed with the NEBuilder^®^ HiFi DNA Assembly Master Mix (NEB, New England Biolabs, Ipswich, MA, USA) following the manufacturer’s instructions. Selected plasmid clones were verified by colony-PCR genotyping and DNA sequencing of the positive amplicons, before being prepared for transformation into *T. atroviride* after PCR linearization.

Oligonucleotides and plasmids used and produced in this study are listed in Supplementary Appendix A.

### 2.4. T. atroviride Transformation

Polyethylen glycol (PEG)-mediated transformation of protoplasts was used as described previously [50] to insert the respective 5.8–6.6 kb large expression or knock-out cassettes—amplified from the respective plasmids by PCR—into the genomes of *T. atroviride* wild type and Δ*tmk1* strains to generate transformants ∆*tmk1*-CRIB and ∆*tmk3*-CRIB, respectively. The intended genetic modifications of the target loci were verified by PCR-based genotyping using locus-specific primer pairs positioned inside and outside of the transformed genome region (Appendix A). Several positive transformant strains of each construct were further validated by DNA sequencing, live-cell imaging and finally purified by three rounds of single spore isolation to reach mitotic stability. The two best clones of each strain were selected for all further studies. Strains used and produced in this study are listed in Appendix A.

### 2.5. Confocal Laser Scanning Microscopy

Stress-free sample preparation was achieved by mounting complete microcolonies of 2–2.5 cm diameter onto large 48 mm by 64 mm glass cover slips (Agar Scientific, UK, cat.# AGL464864-15) using the inverted-agar block method [51] but only cutting around and never across the mycelium. Evaluation of the mycoparasite-prey interaction was performed on a Leica TCS SP5 II (Leica Microsystems, Wetzlar, Germany) confocal microscope using 488 nm excitation light of an Argon laser line and detecting GFP fluorescence between 495–550 nm, selected with an acousto-optic beam-splitter (AOBS), with the in-built HyD detector. Image and time course recording used a Leica HCX PL APO 63x 1.3NA objective lens with laser excitation intensities and scanning times reduced to the minimum required to achieve good signal-to-noise ratios while avoiding phototoxic stress to the cells. 

### 2.6. Wide-Field Fluorescence Microscopy

Wide-field fluorescence microscopy was used to quantify pharmacological GTPase inhibition in liquid germling cultures (see below). Imaging was conducted on a fully motorized Nikon Eclipse Ti2-E microscope base equipped with Nikon CFI Plan-Fluor 40x/0.75 NA and 60X/0.85 NA objectives and an Andor Zyla 5.5sCMOS monochrome camera. Then, 490 nm LED light from a CoolLED p4000 LED unit was used at 5% of the full intensity to excite CRIB-GFP reporter molecules, and emission light between 515–545 nm was collected through a Nikon F66-413 quadband filter cube.

### 2.7. Live-Cell Imaging Experimentation

Imaging data was collected with at least three biological replicates per mycoparasite-prey interaction per imaging session and was experimentally repeated at least twice. Furthermore, individual samples for pre-contact and contact stages were used to ensure that imaging of the pre-contact interaction did not negatively influence hyphal behaviour in the subsequent contact stage. Hence, for each mycoparasite-prey interaction, at least 18 imaging samples were prepared, amounting to a total of at least 108 individual experimental samples for strains of *T. atroviride* CRIB (wt, ∆*tmk1* and ∆*tmk3*), *B. cinerea,* and *F. oxysporum*, plus additional interactions between *T. atroviride* CRIB and *R. microsporus*.

In each of these about 120 experimental samples, at least 10 image recordings, including still Z-stacks and time courses, were taken. Hence, more than 1200 imaging data files are the basis of the presented analyses.

### 2.8. Pharmacologial Inhibition of GTPase Activity

The CRIB reporter binds to both GTPases, Cdc42 and Rac1. Functional differentiation is possible through selective, pharmacological inhibition of GTPase activity. The Cdc42-guanine nucleotide exchange factor (GEF) inhibitor ZCL278 (Tocris, Cat.No. 4794) was used at 100 µM final concentration to selectively block Cdc42 activation, whereas the Rac1-GEF inhibitor NSC23766 (Tocris, Cat.No. 2161) was used at 50 µM final concentration to selectively block Rac1 activation. Notably, ZCL278 is difficultly soluble in water with a strong tendency to precipitate. This decreases bioavailability over time and reduces the actual drug concentration that can act on the cells.

Both drugs were kept as 100 mM stocks at −20 °C and freshly diluted to working concentrations before each experiment. NSC23766 was stored and dissolved in sterile water, whereas ZCL278 was stored in dimethyl sulfoxide (DMSO) and diluted in water.

GTPase inhibition assays were carried out in 8-well-chambered cover slides (ibidi, Cat.No. 80826) using liquid germling cultures of *T. atroviride* conidia in 200 µL/well potato dextrose broth (PDB) diluted to 25% with water in order to reduce background fluorescence of the assay medium. Germlings were pre-cultured for 16–18 h at 25 °C in the dark before imaging.

### 2.9. Image Post-Processing and Automated Time Course Analysis

Image post-processing was performed with Fiji software [52] and restricted to brightness and contrast adjustment, image cropping, and z- or t-projections. Automated quantitative image analyses of the subcellular CRIB reporter dynamics in individual hyphae of *T. atroviride* were performed using segmentation based on local thresholding and mathematical morphology methods. The apical extent and dynamics of the CRIB reporter was quantified using TipTracker version 3, a bespoke local user interface (LUI) software running in a MATLAB environment [30]. The FIJI Image J plugin TrackMate [53], was used to quantify rates of tip growth.

Where possible, quantification of hyphal phenotypes were extracted from the imaging data. Alternatively, the observation frequency of evaluated phenotypes are indicated as percentages.

### 2.10. Manual Image Data Quantification and Statistical Analyses

From our imaging data pool (2.7), five to ten representative imaging data files from each test condition were used for quantitative analysis of particular hyphal phenotypes or cellular effects. Whenever possible, up to 100 individual measurements were extracted from the imaging data, or alternatively, in cases where biological variation was high due to fast tip growth or movement in the Z plane, the number of morphogenetic changes in question were counted and expressed as percentage of the total number of observations. Quantitative measurements were recorded using FIJI ImageJ (https://imagej.net/Fiji accessed on 25 January 2017), graphically processed in Microsoft Excel and statistically evaluated with IBM SPSS software (https://www.ibm.com/analytics/spss-statistics-software accessed on 29 April 2021). Error bars were calculated as standard deviations and statistical significance was determined by *t*-test.

## 3. Results

### 3.1. Dynamic Recruitment of the CRIB Reporter Reflects Apical GTPase Activity in T. atroviride Hyphae

Live-cell imaging confirmed that the CRIB reporter accumulated in the expected crescent shape in 2D in all actively growing tip apices, with the Spitzenkörper standing out at its central subapical position (Figure 1). Upon tip growth arrest, CRIB reporter fluorescence disappeared, but reappeared as soon as tip polarity was re-established and hyphal growth resumed.

The observed recruitment dynamics of the CRIB reporter in *T. atroviride* was identical to that reported for the original *N. crassa* strain indicating its full functionality with respect to capturing the rapid on/off-switching of localised GTPase activity in relation to polarised tip growth in real time. This notion was further confirmed by pharmacologically-induced CRIB dispersal using the Cdc42- and Rac1-specific inhibitors ZCL278 and NSC23766, respectively (Appendix A). To whether CRIB reporter dynamics also reflect the relocation of GTPase activity clusters preceding and thus determining changes in tip growth direction also in response to external signalling cues in *Trichoderma*, self-avoidance responses between leading hyphae of the colony edge were quantified using TipTracker software (Figure 2).

TipTracker analyses confirmed that chemotropic avoidance of individual hyphae strictly follows the lateral displacement of GTPase activity away from the negative stimulus. In other words, the perception of chemorepellent signals first triggers relocation of GTPase activity around the tip apex leading to the reorientation of targeted exocytosis away from the stimulus, which subsequently causes a change in growth direction. Consequently, CRIB reporter dynamics can serve as diagnostic indicator for early chemotropic hyphal-hyphal interactions and provide a tool to evaluate the mutual morphogenetic influences between individual hyphae of the mycoparasite and the potential prey fungus. 

This assumption was confirmed in dual confrontation assays between microcolonies of *T. atroviride* and *B. cinerea*. Live-cell imaging of the subcellular CRIB reporter distribution revealed that hyphae of *T. atroviride* experience significant cell polarity stress already in the pre-contact (i.e., physical pre-contact, but chemical post-contact) interaction stage (Figure 3). Notably, with decreasing distance to the prey, GTPase activity was increasingly delocalised resulting in deformed hyphal tip shapes and tip growth arrest. Only hyphae within the contact zone between both microcolonies and with less than ~1 mm distance to prey hyphae showed signs of induced tip depolarisation. Hyphae outside of this “stress zone” displayed focussed recruitment of apical GTPase activity clusters and normal exploratory tip growth morphology (“exploratory zone”). This strongly suggests that chemorepellent compounds released by *B. cinerea* have an effective range of about 1 mm in the agar medium used.

Notably, despite the severe growth impairment experienced upon first contact, *T. atroviride* eventually overcomes this blockage and successfully mycoparasitises *B. cinerea*. The first indications that prey-induced depolarisation is only a transient effect in this particular pairing becomes apparent in the “adaptation zone”. This is characterized by (i) restored CRIB reporter recruitment, (ii) resumption of polarised tip growth, (iii) multi-polarisation, and (iv) the establishment of physical contact to prey hyphae.

The effective distance and severity of polarity stress imposed varied slightly depending on the prey species confronted. The overall phenotype, including CRIB reporter dispersal and tip growth arrest, was nevertheless conserved in all tested interactions.

To better understand the cellular processes that occur at the transition from exploratory zone to stress zone, we quantified the relation between CRIB reporter recruitment and hyphal tip growth speed.

### 3.2. The Onset of Prey-Induced Tip Depolarisation during the Pre-Contact Phase Occurs Abruptly and Results in Growth Arrest

The intensity of the fluorescent CRIB reporter signal reflects GTPase activity and is thus proportional to hyphal tip growth speed. We hypothesised that sensing of prey-derived chemorepellents will gradually slow down exploratory growth and probably convert it into hyphal avoidance growth in a dose- and distance-dependent manner. Surprisingly though, the effect of prey-induced tip depolarisation was in all investigated cases very sudden (Figure 4). The dispersal of the apical CRIB signal led to the immediate cessation of polarised tip growth and, in some cases, slight tip bulging. This is likely the result of residual, non-focused GTPase activity and temporal continuation of exocytosis at the tip apex. 

One important difference this analyses also revealed is that prey-induced tip depolarisation can be transient or permanent, and this is directly correlated to the known macroscopic outcome of each particular interaction [8]. Similar as with *B. cinerea*, *T. atroviride* can overcome the defence responses of *R. microsporus* eventually, and thus successfully mycoparasitise this prey, defining both species as “easy prey”. In interaction with *F. oxysporum*, however, prey-induced tip growth arrest persists permanently and *T. atroviride* cannot adapt to the stress and mycoparasitise this species, defining *F. oxysporum* as “difficult to impossible prey”; under the given experimental conditions (Appendix A).

### 3.3. Mycoparasitic Hyphae Switch Chemotropic Behaviour While Establishing Contact to Prey Hyphae

In interactions with easy preys, growth-arrested hyphae of *T. atroviride* eventually manage to repolarise and resume growth into the prey mycelium. One noticeable pattern during this regrowth is the directed approach to prey hyphae resulting in contact (Figure 5). In part, this results from spatial confinement due to the increasing density of prey hyphae occupying the same space (“hyphal crowding”). In addition, however, it may also result from chemotropic and thigmotropic hyphal-hyphal interactions. Three morphological features were consistently observed: wavy growth along prey hyphae, contact-induced multi-polarisation to give a local, densely branched attack structure, and the maintenance of close contact to prey hyphae.

Hyphae of *T. atroviride* were observed with a frequency of 75% to be attracted by individual prey hyphae up to the point of physical contact. Contact establishment often occurred at a single point before the hyphal tip was repelled by some other chemical or tactile trigger. This approach-and-repel phenotype was repeated several times, leading to a wavy growth pattern of the *T. atroviride* hyphae (Figure 5). Compared to the more frequently observed “growing along” prey hyphae without waving movements, this specific phenotype occurred at a frequency of about 25% in the interaction with hyphae of *B. cinerea* (Appendix A). Quantification of CRIB reporter intensities and dynamics showed that localised GTPase activity correlated directly with subsequent changes in tip growth direction and velocity. We infer that physical contact to prey hyphae cannot simply be established anywhere, but that hyphae of *T. atroviride* have to find the right local conditions to overcome chemical and probably tactile barriers presented in close proximity to and on the surface of prey hyphae.

The second striking morphological feature frequently observed in the contact phase was multi-polarisation of mycoparasitic hyphae to give a local highly branched attack structure (Figure 6). This phenomenon occurred in very close proximity and upon physical contact with prey hyphae, and is likely the consequence of the disruption of a unipolar growth axis and its subsequent restoration as multiple tip growth axes.

Contact-induced multi-polarisation was previously observed in interactions between *T. atroviride* and *Phytium ultimum* [20] and suggests that *Trichoderma* exploits the contact signal to adapt its growth morphology to maximise its mycoparasitic activity against particular prey species. This finding has significant implications because it represents another signalling-based microparasitic phenotype that correlates with the macroscopic outcome of mycoparasite-prey interactions in a prey-specific way.

In case neither of these two specific morphological responses were triggered, hyphae of *T. atroviride* tend to simply grow alongside prey hyphae without obvious signs of chemical or tactile interaction. Whether subtle changes in CRIB reporter dynamics may help to differentiate this neutral growth behaviour from simple growth along an inert physical barrier need to be resolved in future investigations. 

The next step, therefore, was to use these two specific microparasitic phenotypes of contact-induced wavy growth and multi-polarisation as diagnostic tools to start dissecting the underlying MAPK/SAPK signalling pathways.

### 3.4. Tmk1 Is Required to Establish Contact with Prey Hyphae

To evaluate the functional relationship between GTPase activity and MAPK/SAPK signalling pathways, *T. atroviride* Δ*tmk1* and Δ*tmk3* gene deletion strains expressing the CRIB reporter were generated and used to analyse hyphal behaviour in confrontation assays. As previously shown [32], colony extension of Δ*tmk1* was decreased in comparison to the parental strain, and prey confrontation created a noticeable persistent gap between the colonies in the interaction with *B. cinerea.* Macroscopic comparison of the confrontation behaviour between both microcolonies confirmed that the extensive tip growth arrest responsible for gap formation became established in *T. atroviride* at the early pre-contact stage (Appendix A). Microscopic analyses of leading hyphae forming the colony margin revealed that there was a higher probability of avoidance responses in ∆*tmk1* than in the wild type (Appendix A), indicating an increased sensitivity to prey defence responses in this mutant. This subsequently resulted in fewer exploratory hyphae invading the prey mycelium and establishing effective contact. Nevertheless, observation of the few exploratory hyphae of ∆*tmk1* that did manage to invade the prey mycelium showed that ∆*tmk1* hyphae quickly loose apical GTPase activity and arrest tip growth upon contact with hyphae of *B. cinerea* (in more than 90% of observations) (Figure 7).

Secondly, in confrontations with *F. oxysporum*, which follow similar dynamics during pre-contact, hyphae of *T. atroviride* ∆*tmk1* displayed avoidance responses in close proximity to prey hyphae in most cases (in 80% of observations). Hence, physical contact to prey hyphae was rarely established at all, in contrast to the wild type and in some instances the Δ*tmk3* mutant when contact was achieved (Figure 8). We infer that pre-contact, negative chemotropic sensing of prey-derived compounds still operates in Δ*tmk1*, but that positive chemotropic responses required to elicit mycoparasitic interaction are compromised. In cases where individual hyphae of *T. atroviride* managed to invade the colony periphery of *F. oxysporum*, contact with prey hyphae induced immediate tip growth arrest in the mycoparasite as seen before in the interaction with *B. cinerea* (Figure 7).

### 3.5. Loss of Tmk3 Reduces Hyphal Survival during Pre-Contact and Impairs the Formation of Effective GTPase Activity Clusters

Genetic deletion of *tmk3* resulted in generally reduced colony extension and specifically early tip growth arrest in the pre-contact phase when compared to Δ*tmk1* (Appendix A). To confirm this apparently increased sensitivity of ∆*tmk3*-CRIB towards prey defence chemicals in more detail we repeated confrontations with *B. cinerea* as a more susceptible prey than *F. oxysporum*. Even in confrontations with *B. cinerea*, hyphae of ∆*tkmk3* were severely affected already in the pre-contact phase of the interaction (Figure 9A). The majority of leading hyphae showed clear signs of impaired polarised tip growth and even cell lysis. The number of damaged hyphae of Δ*tmk3* is massively increased in comparison to the wild type, whereas the number of healthy hyphae is significantly decreased in the mutant (Appendix A). We infer that the lack of SAPK signalling renders the mutant highly sensitive to prey-derived defence compounds. Exploratory hyphae growing beyond a “safe distance” from *B. cinerea* showed neither growth impairment, nor signs of prey-induced polarity stress, evident by apically focused CRIB reporter recruitment (Figure 9B).

Interestingly, exploratory hypha in Δ*tmk3*-CRIB that overcame prey-induced polarity stress and managed to invade the prey mycelium suffered severe polarized growth impairment on approach to prey hyphae (Figure 9C). In a similar manner to WT-CRIB (Figure 5), contact search behaviour along prey hyphae was occasionally observed in Δ*tmk3*-CRIB, but maintenance of unipolar growth axes that was still possible in Δ*tmk1*-CRIB (Figure 7) was not observed. Instead, CRIB reporter fluorescence was dramatically fragmented and dislocated into several GTPase activity clusters preventing the establishment of functional mycoparasitic interactions (Figure 10).

As expected, this developmental delay was even more pronounced when *T. atroviride* ∆*tmk3* was confronted with a more difficult fungal prey, such as *F. oxysporum*, leading to growth arrest in the pre-contact and contact phases sooner, and eventually preventing the establishment of mycoparasitic overgrowth at the macroscopic level (Appendix A). 

## 4. Discussion

### 4.1. CRIB Reporters Facilitate Quantification of Hyphal Morphogenesis in Mycoparasitic Interactions

The Rho-type GTPases Cdc42 and Rac1 are important key players in the regulation of eukaryotic cell polarity [22,25,54]. Localised recruitment of active-state, GTP-bound GTPases at sites of polarised cell growth was initially demonstrated using fluorescent CRIB reporters in yeast [29,55], and subsequently in *Candida albicans* [56] and *Neurospora crassa* [30]. In the latter, CRIB reporter technology was instrumental in dissecting unique and overlapping functions of the highly homologous CDC42 and RAC1 proteins in germling and hyphal morphogenesis, cell-cell fusion, and MAPK signalling.

In this study we generated CRIB reporters for the mycoparasite *T. atroviride* based on the PBD of its PAK Cla4, following the approach initially used for *N. crassa* [30], but at the same time improving the photophysical properties of the fluorescent tag. Modification of sGFP (S65T) [14,57] to the new version we termed mBasicGFP (F64L, S65T, S72A, and A206K) gave significantly enhanced subcellular visibility and spatial resolution of localised GTPase activity clusters, and thereby considerably improved automated quantitative image analyses using TipTracker [30]. 

CRIB-mBasicGFP fluorescence was localised at actively growing hyphal tips in a typical crescent (in 2D) or cap shape (in 3D), as well as in the subapical Spitzenkörper, and disappeared shortly after tip growth arrest. Furthermore, TipTracker analyses verified that repositioning of GTPase activity clusters along the apical dome preceded changes in growth direction in response to chemotropic cues in a similar manner to *N. crassa* [30]. Together, these findings confirmed that the CRIB reporter designed for *T. atroviride* highlights the 4D dynamics of Cdc42 and Rac1 activity consistent with the original CRIB reporter in *N. crassa* [30], and provides a useful tool to investigate the regulation of hyphal morphogenesis in mycoparasitic interactions.

### 4.2. T. atroviride Has to Overcome Prey-Induced Polarity Stress to Initiate Mycoparasitic Morphogenesis

The first observation from CRIB reporter analyses was the considerable polarity stress experienced by *T. atroviride* during prey encounter, even in interactions that would subsequently lead to mycoparasitism. We note that considerable extension was given to injury-free preparation of all microscopy samples to ensure that any observed stress phenotypes resulted from the interaction between mycoparasite and prey fungus, and not from handling artefacts. The second revealing insight from these analyses was that the ability to overcome the stress situation was a major determinant for the outcome of the mycoparasitic interaction on the colony level. Confronted with *B. cinerea*, *T. atroviride* adapted and was able to resume polarised growth, eventually leading to successful mycoparasitism of its prey. In contrast, when confronted with *F. oxysporum*, hyphae of *T. atroviride* were permanently growth arrested in the pre- and early-contact phases, and thus could not effectively invade and mycoparasitise the opposing mycelium. Using these dynamics as rating scheme we propose to classify fungi as easy, difficult, or impossible preys for *T. atroviride*.

Although it has been previously reported that putative prey fungi respond to the mycoparasitic attack with secondary metabolite, enzyme, and ROS secretion for their defence [58,59,60], the observed morphological impact on hyphae of the mycoparasite were unexpectedly severe. The initial tip bulging phenotype resulted from temporarily continued exocytosis after apical GTPase activity clusters were disrupted. When these clusters were not restored, the functional chain of GTPase activity, F-actin polymerisation, vesicle delivery, and targeted exocytosis [25] collapsed and polarised tip growth arrested completely. The ability to restore this functional chain determines whether or not the putative prey can be eventually mycoparasitised.

The outcome of mycoparasitic interactions, of course, also depends on additional conditions. For instance, we recently showed that the ability of *T. atroviride* to mycoparasitise *F. oxysporum* is light regulated and successful under yellow or red light as well as in complete darkness [42]. One possible explanation for this phenomenon is that the light-dependent mycotoxin production of *F. oxysporum* [61] is so significantly reduced that *T. atroviride* can overcome its defence response. This example shows that it will be increasingly important to monitor additional experimental factors during investigation of fungal-fungal interactions rather than just using standard agar medium, fixed temperature, and light on or off conditions. Substrate texture, nutrient heterogeneity, C/N-ratios, quantity and quality of light, temperature regimes, and numerous other features may need modifications to create an environment closer to natural conditions. 

The next set of observations in description of the mycoparasitic process at the level of individual hyphae were based on quantifying CRIB reporter dynamics in hyphal-hyphal interactions. Mycoparasitic hyphae that successfully invaded the prey mycelium displayed contact search behaviour, characterised by rapid switching between chemoattraction and chemorepulsion, likely in combination with the perception and response to cell surface cues or extracellular vesicles as a possible route for the transient delivery of signals, toxins, or other types of effector molecules as reported in mycorrhizae and pathogens [62,63,64]. This data suggests that the establishment of physical contact with prey hyphae cannot simply occur anywhere, but that certain chemical and tactile features on the surface of prey hyphae must be suitable. Earlier studies suggest a prominent role for lectins in this process [65,66,67]. Interestingly, hyphal contact between *T. harzianum* and *R. solani* was mediated by an external galactose rich matrix produced by the putative prey and lectins produced by *Trichoderma* [68], whereas other reports indicated that galactose present in the cell wall of the mycoparasite interact with lectins from the prey [66,69,70]. We suspect that locating such features is a key function of exploratory hyphae which upon detection send signals back into the colony (sub-)periphery to initiate further invasion of the prey mycelium. In case where such positive signals are not generated above a certain threshold, perhaps because polarised tip growth of exploratory hyphae is compromised, the mycoparasitic attack remains blocked.

Two contact-induced morphogenetic events stand out in this context: multi-polarisation and contact maintenance. Both features have been described earlier and functionally attributed to mycoparasitic attack [20]. In light of the improved visualisation technologies at our disposal, it seems appropriate to re-evaluate cause, effect, and function of these structures. 

Our comparative analysis, furthermore, indicates that contact angle and duration of contact to prey hyphae may influence morphogenetic changes required for mycoparasitism. Multi-polarisation was most frequently (in at least 70% of observations) observed when exploratory hyphae hit prey hyphae at steep contact angles. Tip growth velocity peaked immediately upon contact and rapidly decreased as soon as the unipolar growth axis split into several tips. The newly formed tips tended to maintain the original growth trajectory, thereby expanding and increasing the local surface area of the invading hyphae. Notably, contact-induced multi-polarisation was established next to the set growth axis and not by lateral branching of the sub apex which is often seen when hyphae encounter a physical barrier. In the latter scenario, physical impact destroys the polarized tip growth apparatus and a new one is formed laterally to the original to rapidly divert the continuous supply of secretory vesicles produced in the expanding parental hyphae to the nearest unrestricted site on the apex. The differences in kinetics between multi-polarisation and branching are, hence, somewhat similar to those known from apical and lateral branching under solitary growth conditions when *Trichoderma* is not confronted by another fungal colony [71,72]. How the vectorial supply of vesicles to the hyphal tip determines thigmotropism in relation to the contact angle has been shown in *N. crassa* [73]. The functional relationship between contact-induced asymmetric distribution of the apical polarity complex and asymmetric hyphal morphologies has been described in *C. albicans* [74]. It is reasonable to propose that the underlying regulatory pathways act in very similar ways to control tip polarity in *Trichoderma*, but further studies on the molecular details of these responses are needed to verify this.

In contrast, at shallow contact angles, hyphae of *T. atroviride* tended to maintain close physical contact by growing along the prey hypha with decreased tip velocity (compared to free growth). From this maintained physical interaction, other specific growth patterns, such as hyphal coiling or the formation of infection structures, may develop in particular mycoparasite-prey pairings at later stages of the interaction [4,75,76]. We conclude that multi-polarisation and the ability to establish and maintain close physical contact to prey hyphae thus provide key morphogenetic signals to switch into a growth mode that facilitates mycoparasitic overgrowth of the adjacent prey mycelium over the course of the next hours and days. This change in microparasitic growth behaviour on the level of the individual hypha therefore ultimately determines the macroparasitic outcome of the interaction on the level of the whole colony. 

### 4.3. The MAP/SAP Kinases Are Required for Polarity Stress Adaptation and Contact-Induced Mycoparasitism

Our previous studies showed that deletion of *tmk1* and *tmk3* significantly reduced the mycoparasitic abilities of *T. atroviride* in comparison to the wild-type strain [32,42]. By analogy to the role of the *F. oxysporum* homologue, *Fusarium* MAPK 1 (Fmk1), in plant sensing [77], Tmk1 is suspected to be governed by pheromone receptors to sense prey-derived signals. However, the precise function of Tmk3 in the mycoparasitic process is still unknown. Previous studies showed that Tmk3/Hog1 of *T. harzianum* becomes active in antagonistic interactions with fungal preys [78], as well as in the presence of fungivorous insects [79]. In yeast, activation of Hog1 is mediated through the transmembrane protein synthetic high osmolarity-sensitive 1 (Sho1) which recruits the MAPK to cortical GTPase/PAK activity clusters in response to stress [25,80,81]. A similar role for Tmk3 in *Trichoderma* may be suggested. Whether or not the third MAPK, Tmk2, is essential for the virulence of *T. atroviride* is still unclear, due to the current lack of a homokaryotic *tmk2* gene deletion strain. Ongoing work attempts to verify this in genotypically distinct mutants.

MAP kinases are known or suspected to be involved in the integration of chemotropic, thigmotropic, and stress signals [82,83]. To test the role of these MAP kinase pathways, we investigated microparasitic interactions of respective MAPK mutants. Our previous studies showed that, although the genetic deletion of the *tmk1* MAPK gene significantly impaired the mycoparasitic activity of *T. atroviride*, the morphological changes typically associated with mycoparasitism, including attachment and growth alongside prey hyphae, apparently remained unaffected [32,33]. The cellular basis for the reduced virulence in the mutant strains thus remained largely unexplained. For the recently generated ∆*tmk3* mutant [42], this is the first study to investigate the cellular basis of its avirulence. Our analyses suggest that the reduction in mycoparasitic virulence of ∆*tmk3* and ∆*tmk1* are quantitative effects, resulting from the reduced abilities to restore polarised tip growth upon prey-induced depolarisation, and to establish contact to prey hyphae, respectively.

The Δ*tmk1*-CRIB deletion mutant displayed slower growth rates in comparison to the WT-CRIB control strain, which not only decreased colony extension speed by 15–20% as shown previously [32], but more importantly provided more time for the putative prey to defend itself effectively. This made ∆*tmk1* more susceptible to prey-induced depolarisation, ultimately leading to increased avoidance and fewer exploratory hyphae reaching the prey mycelium. The few hyphae that do, are either chemotropically repelled before they come into physical contact to prey hyphae or they are likely to be growth arrested when they do. As this inhibiting effect is already established in the pre-contact phase and reinforced in the early-contact phase it massively reduces the number of hyphae that could generate morphogenetic signals required to launch a proper mycoparasitic attack. The macroscopic consequence of this microparasitic inhibition is retardation of the mycoparasitic development of the ∆*tmk1* mycelium by several days in comparison to the wild-type strain. It also explains the previously observed host-specificity, i.e., the observation that *T. atroviride* ∆*tmk1* was able to eventually mycoparasitise *R. solani,* but not *B. cinerea* [32]. In the light of the new data, we suspect that this is associated with a weaker defence response elicited by *R. solani* allowing better morphogenetic adaptation of the mycoparasite. Fus3 in *S. cerevisiae*, a homolog of Tmk1, promotes polarised growth by inducing F-actin polymerisation through the phosphorylation of the formin bud neck involved 1 (Bni1) [84,85,86], and via down-regulation of the Rho-GTPase activating protein (GAP) Rga2 [87]. Hence, the loss of Tmk1-mediated positive feedback on GTPase activity in *T. atroviride* could explain the inability of ∆*tmk1* hyphae to properly engage in physical interaction with prey hyphae, despite being able to chemotropically sense their presence.

Genetic deletion of *tmk3* resulted in a generally similar inhibition of the macroparasitic outcome, albeit the microparasitic effect is slightly more severe compared to ∆*tmk1*. The majority of peripheral hyphae in the ∆*tmk3* strain are much more heavily affected by prey-induced cellular damage during the pre-contact phase. The most likely explanation is the lack of Tmk3-mediated SAPK signalling for stress adaptation [40], which—as for ∆*tmk1*—results in even fewer exploratory hyphae reaching hyphae of the prey. However, in contrast to ∆*tmk1*, the few hyphae of ∆*tmk3* that do invade the prey mycelium suffer pronounced disruption of unipolar tip growth. Quantitative CRIB analyses revealed that focused GTPase activity clusters can no longer be established, but are instead randomly dispersed along the cell apex leading to the characteristic corrupted phenotype known from typical cell polarity mutants [30,88,89,90,91]. The induced polarity defect makes it impossible for the mutant to establish functional mycoparasitic contacts and renders it avirulent as previously noted [42]. A direct link between stress signalling via the Hog1-ortholog SAPK suppressor of tyrosine phosphatase (Sty1) and Cdc42-dependent cell polarity has recently been established in fission yeast using CRIB reporters [92]. The study showed that Sty1 activation in response to stress is sufficient to disperse GTPase activity cluster from the cell apex and terminate cell elongation. This suggests that inhibition of the SAPK pathway translates into loss of control over focused GTPase activity [93]. This offers an explanation why in the absence of Tmk3-mediated stress regulation, apical GTPase activity is maintained in *T. atroviride*, albeit in a highly deregulated manner. In conclusion, the interplay between SAPK and GTPase signalling is essential to establish functional mycoparasitism in *Trichoderma*.

## Figures and Tables

**Figure 1 jof-07-00365-f001:**
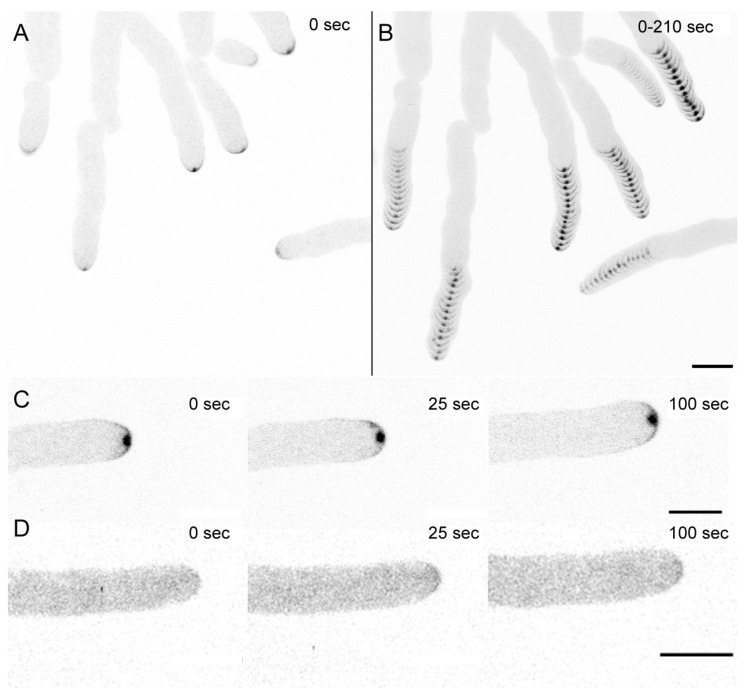
Functional localization dynamics of the *T. atroviride* CRIB reporter. (**A**) single time point image and (**B**) time course projection of CRIB-mBasicGFP dynamics in growing hyphae showing consistent recruitment of the CRIB reporter as apical crescent and as part of the subapical Spitzenkörper (Spk, arrowhead). Scale bar, 10 µm. The time course sequence is provided as Appendix A. (**C**) time course showing continuous CRIB reporter recruitment during active tip growth and (**D**) complete absence of tip localised fluorescence in the wild type control expressing no CRIB reporter. Scale bar, 5 µm.

**Figure 2 jof-07-00365-f002:**
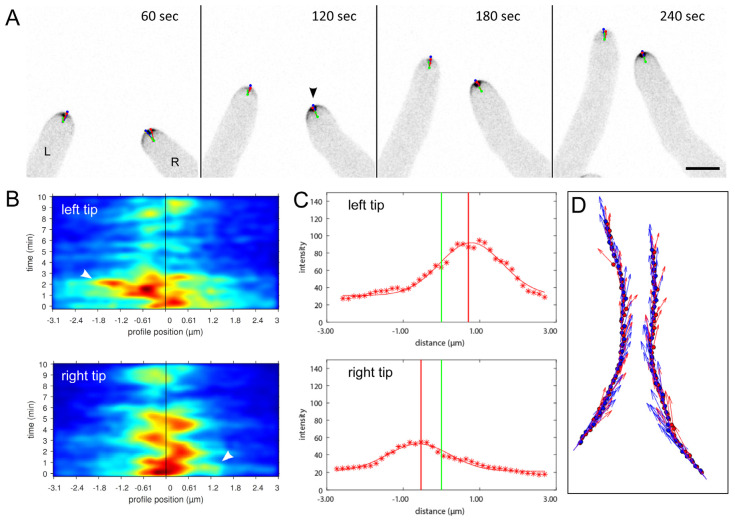
Lateral displacement of GTPase activity clusters at the tip apex controls directional growth during hyphal self-avoidance responses. (**A**) Time course of dynamic relocation of CRIB reporter fluorescence during negative autotrophy (chemotropic hyphal avoidance). Essentially, CRIB reporter fluorescence, and therefore GTPase activity, relocates away from adjacent hypha before changes in tip growth direction are morphologically realised. The point of the highest fluorescence signal (red dot) within the tip apex is identified as the maximum of a Gaussian fit to the intensity distribution of CRIB reporter fluorescence. The movement of the CRIB vector (red vector) shows the lateral repositioning of GTPase activity clusters at the plasma membrane exiting from the centre of the tip (green dot). Blue dots mark the point of maximum curvature, i.e., the actual tip of the hypha, and movement of the apex vector (blue vector) indicates a directional change in polarized tip extension. Scale bar, 5 µm. The time course sequence is provided as Appendix A. (**B**) The corresponding heat map visualizes the intensity distribution of CRIB reporter fluorescence around the tip apex, and accentuates the fluctuating displacement of the apical crescent from the apex centre (black vertical line). The rapid reorientation of the left hypha (L) towards the left, i.e., away from the hypha on the right, becomes all the more obvious. Likewise, the point of maximum reorientation of the right hypha (R) towards the right (arrowhead), corresponds to the arrowhead in image (**A**). (**C**) Signal displacement plots show the distance of the fluorescence maximum (red line) relative to the tip centre (green line) and emphasize the unilateral displacement of GTPase activity in each tip. (**D**) The trajectories of the CRIB intensity vector (red arrow) and tip apex vector (blue arrow) show that GTPase activity is displaced prior to reorientation of tip growth. The general observation in all cases is, that GTPase activity relocates before the directional change in tip growth follows.

**Figure 3 jof-07-00365-f003:**
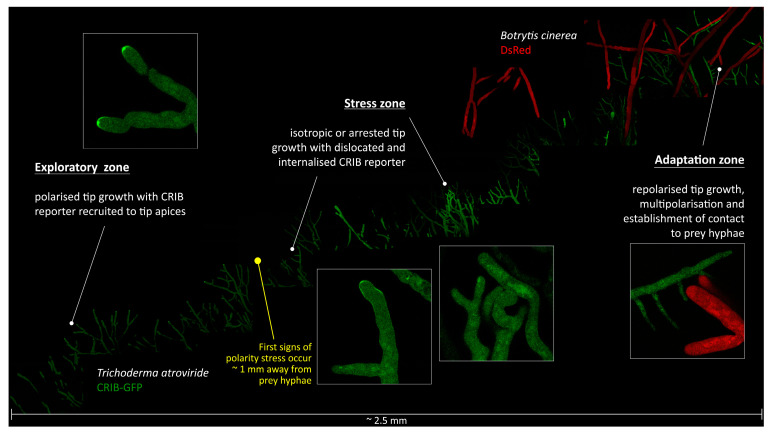
The morphogenetic effects of *B. cinerea* on *T. atroviride* expressing the CRIB reporter reveal significant polarity stress resulting from defence responses elicited by the prey fungus. Composite LSCM image covering the no-contact, pre-contact, and first-contact regions between *T. atroviride* (CRIB-GFP, green) and *B. cinerea* (cytoplasmic DsRed). With respect to the morphogenetic changes of the mycoparasite, evident by the dislocation of the CRIB reporter, the 2.5 mm wide area can be divided into three zones: the exploratory zone, characterized by normal tip growth morphology and focussed recruitment of active GTPases into the tip apex; the stress zone, represented by irregular dispersal of GTPase activity along the apex and accumulation of CRIB reporter fluorescence inside what appear to be degrading vacuoles; and the adaptation zone, in which hyphae manage to restore tip polarity and establish physical contact to prey hyphae. Appendix A shows another example of the extensive prey-induced depolarisation phenotype of mycoparasitic hyphae.

**Figure 4 jof-07-00365-f004:**
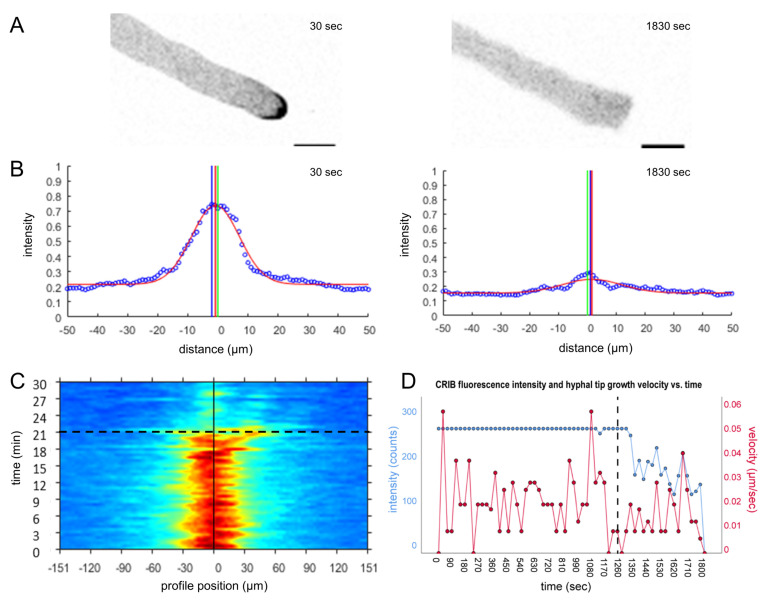
CRIB reporter dispersal and concomitant tip growth arrest occur within seconds. 31-min time course (1 frame every 30 s) of *T. atroviride* WT-CRIB hyphae in the pre-contact zone around *B. cinerea*. (**A**) Focussed GTPase activity during polarised tip growth with an initial average growth speed of 0.02 µm/s. Upon the perception of prey-derived polarity-stress signals, apical GTPase activity and tip growth speed declined rapidly. Scale bar, 10 µm. The time course sequence is provided as Appendix A. (**B**) CRIB reporter signal displacement plot at t = 30 and 1830 s (30 min) of the time course, showing that the hypha in (**A**) is growing with high GTPase activity (70% signal intensity). Seconds later, the CRIB reporter signal has dropped to background level (20% signal intensity) and the hyphal tip loses its polarised shape. (**C**) The corresponding heat map visualises the sudden shutdown of GTPase activity. (**D**) CRIB signal intensity and tip velocity plot displaying the sudden drop in tip growth speed occurring almost simultaneously after the cessation of apical GTPase activity. The black dotted line indicates that time point in (**C**,**D**).

**Figure 5 jof-07-00365-f005:**
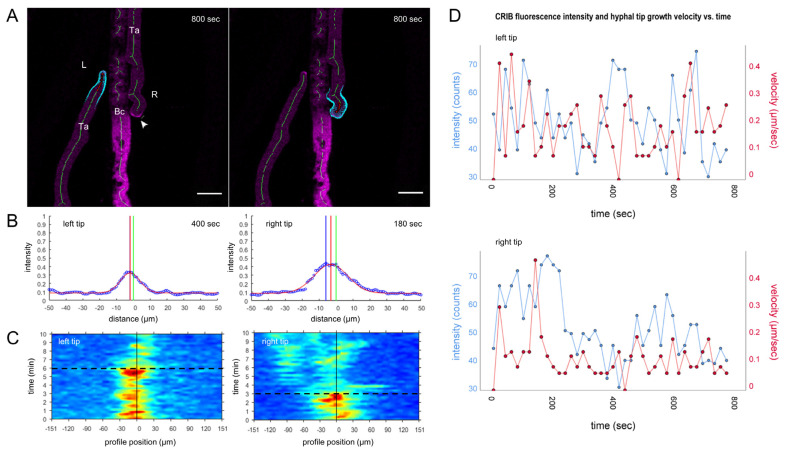
To establish physical contact to prey hyphae *T. atroviride* has to overcome chemical and tactile barriers presented by *B. cinerea*. (**A**) Two hyphae of *T. atroviride* WT-CRIB approach a hypha of *B. cinerea* (Bc). The switching between positive and negative chemotropic growth is clearly reflected by changing CRIB reporter dynamics, including the simultaneous formation of two apical GTPase activity clusters oriented in opposite directions (arrowhead). Scale bar, 5 µm. The time course sequence is provided as Appendix A. (**B**) Displacement plots of CRIB fluorescence relative to the tip centre emphasize that the highest GTPase activity is oriented towards the respective growth direction of each hypha at t = 180 and 400 s. (**C**) The heat map accentuates the fluctuating dislocation of apical GTPase activity from the apex centre (black vertical line) over the time course, and illustrates that GTPase activity eventually disperses in the right tip, suggesting that the attachment attempt ended unsuccessfully. The black dotted line indicates the time point related with time displacement plots presented in (**B**). (**D**) Maximum CRIB fluorescence signal and hyphal tip growth velocity plotted against time, showing that GTPase activity peaked at around 400 s (left tip) and at around 180 s (right tip), correlating with the signal peaks in both displacement plots (**B**) and heat maps (**C**). Changes of tip growth velocity correlate with fluctuations of CRIB intensity over time. The left tip is approaching the prey hypha with higher velocity and just initiated the contact process.

**Figure 6 jof-07-00365-f006:**
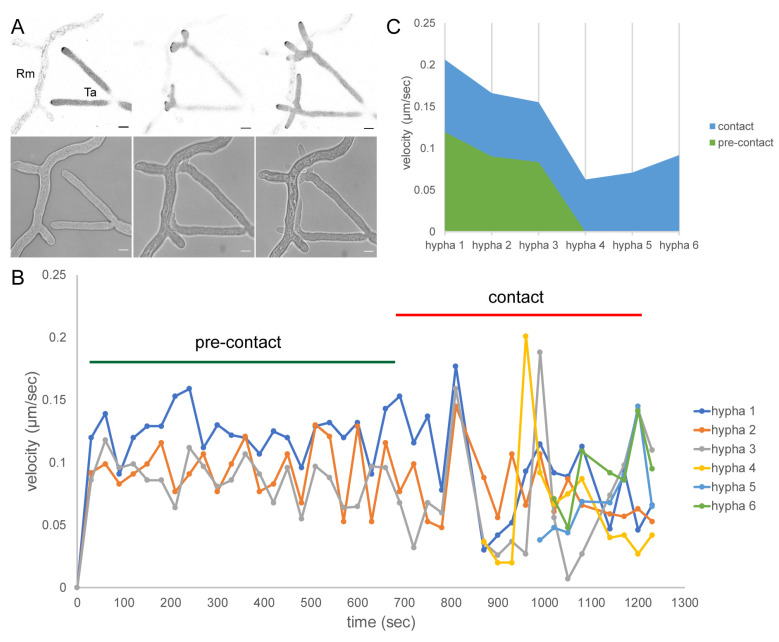
Contact-induced multi-polarisation of mycoparasitic hyphae. (**A**) Two individual hyphae of *T. atroviride* WT-CRIB (Ta) approach a hypha of *R. microsporus* (Rm). The apical CRIB reporter localisation clearly shows that upon contact to the prey hypha the previously unipolar growth axis splits into multiple growth axes, most evident by the formation of several Spitzenkörper which drive the protrusion of individual hyphae off the splitting point. Scale bar, 10 µm. The time course sequence is provided as Appendix A. (**B**) Hyphal tip growth speed plotted against time (t = 20 min), showing that GTPase activity correlates with changes in tip velocities once contact with the prey hypha is established. (**C**) Representation of the rapid change in tip velocity during pre-contact and contact. Upon contact, tip velocities of hyphae 1, 2, and 3 initially peak and then decrease by 28%, 16%, and 14%, respectively, as the new tips commence growth.

**Figure 7 jof-07-00365-f007:**
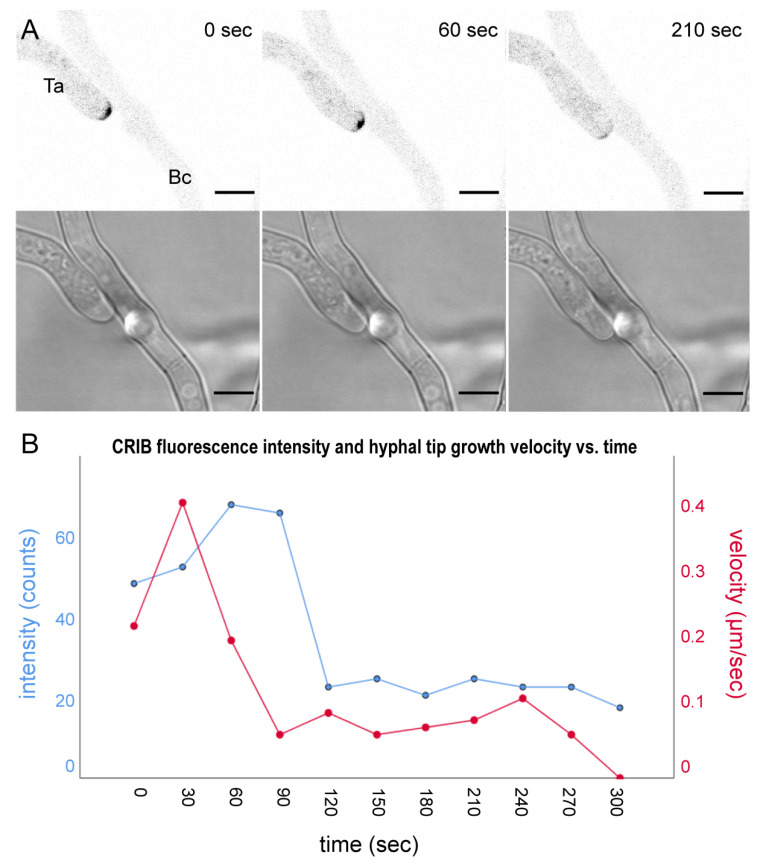
GTPase activity and tip growth arrest upon prey contact in *T. atroviride* Δ*tmk1*-CRIB. (**A**) Time course montage of the CRIB signal and corresponding bright field image of *T. atroviride* ∆*tmk1*-CRIB mutant (Ta) coming into contact with *B. cinerea* (Bc). The mutant was attracted by prey hyphae and two minutes after physical contact with prey hyphae the CRIB signal disappeared, and tip growth arrested. Scale bar, 5 µm. The time course sequence is provided as Appendix A. (**B**) CRIB reporter signal intensity and tip growth speed, plotted against time. As apical GTPase activity decreased, hyphal tip growth speed declined.

**Figure 8 jof-07-00365-f008:**
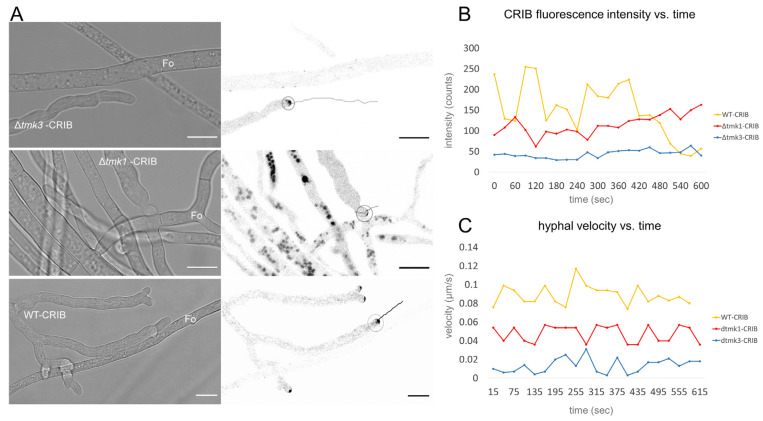
Chemotropic growth responses of *T. atroviride* ∆*tmk1*-CRIB, Δ*tmk3*-CRIB, and WT-CRIB towards *F. oxysporum*. (**A**) Brightfield and inverted fluorescence images showing growth responses and subcellular CRIB reporter dynamics, respectively, of hyphae of ∆*tmk1*-CRIB, Δ*tmk3*-CRIB, and WT-CRIB strains in confrontation with hyphae of *F. oxysporum* (Fo). Hyphae of ∆*tmk1*-CRIB predominantly displayed chemotropic avoidance in close proximity to hyphae of *F. oxysporum* and did not establish lasting physical contact, whereas wild type and ∆*tmk3* strains were attracted to and continued to grow along prey hyphae upon physical contact. Scale bar, 10 µm. The time course sequences are provided as Appendix A. (**B**,**C**) Interestingly, CRIB signal intensity and dynamics varied noticeably between the three strains, but remained functionally linked in each scenario. Once the CRIB signal increased or decreased it was followed by corresponding increase or decrease of hyphal tip growth velocity, respectively. Overall, ∆*tmk1*-CRIB and Δ*tmk3*-CRIB strains exhibited lower CRIB intensity and velocity profiles in comparison to WT-CRIB strain, with ∆*tmk1*-CRIB always being stronger and faster than ∆*tmk3*-CRIB.

**Figure 9 jof-07-00365-f009:**
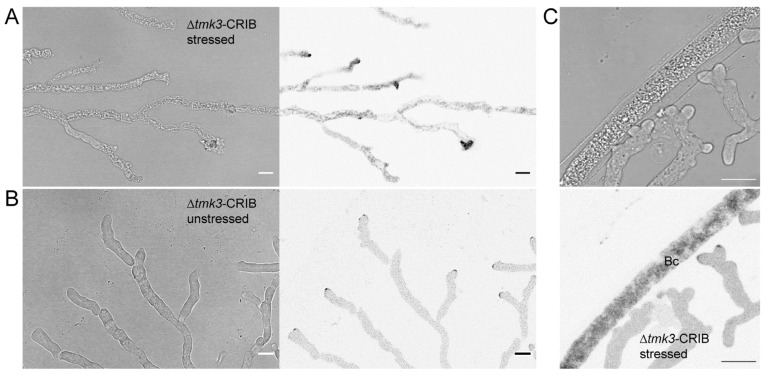
Cellular damage induced by pre-contact interaction with *B. cinerea* in *T. atroviride* ∆*tmk3*-CRIB hyphae. (**A**) Leading hyphae of *T. atroviride* ∆*tmk3*-CRIB suffered from cell lysis and impaired polarised growth when confronted with compounds released by *B. cinerea* into the culture medium. Scale bar, 10 µm. (**B**) Leading hyphae growing beyond a “safe distance” from the prey fungus showed normal, unstressed growth morphology and functional recruitment of the CRIB reporter. Scale bar, 10 µm. (**C**) Hyphae in close proximity to fungal prey hyphae may attempt to establish contact but suffered from severe disruption of unipolarised tip growth leading to multiple bulged hyphal tips. Scale bar, 10 µm.

**Figure 10 jof-07-00365-f010:**
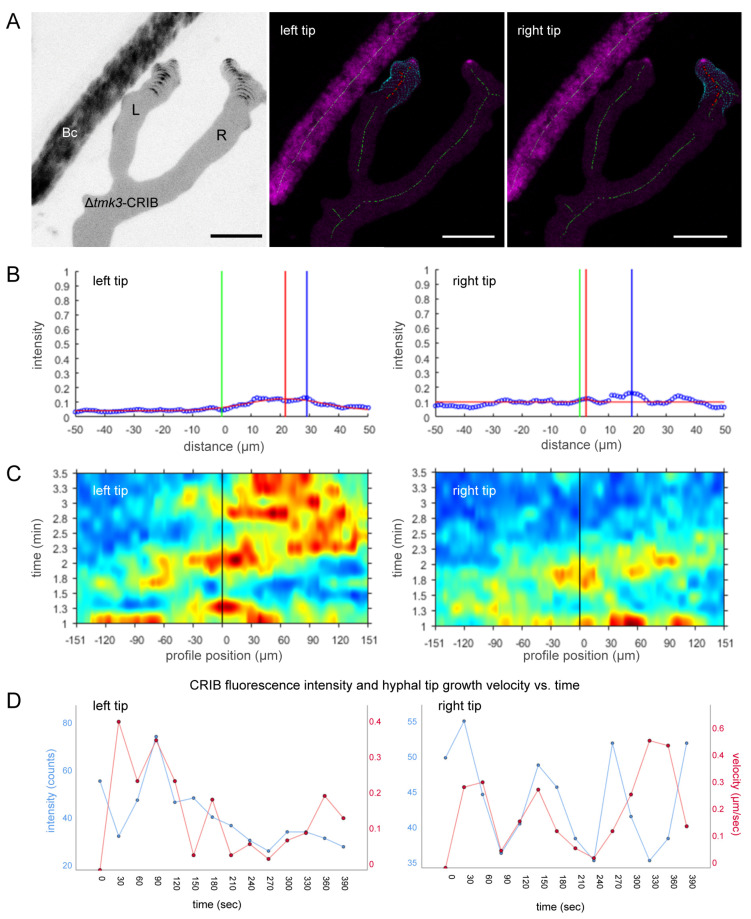
Lack of Tmk3 disrupts the formation of focussed GTPase activity clusters. (**A**) Apical recruitment of the CRIB reporter was restricted to an unusually small region of the hyphal tip, indicating that the formation of functional GTPase activity clusters was significantly impaired. Although positive chemotropic interaction with prey hyphae appeared possible, establishment of physical contact occurred with greatly reduced efficiency. Scale bar, 10 µm. The time course sequence is provided as Appendix A. (**B**) Displacement plots demonstrate that apical CRIB intensity levels were significantly lower and less focussed in the Δ*tmk3*-CRIB mutant compared to the *T. atroviride* WT-CRIB (see Figure 5B). (**C**) The corresponding heat map to (**A**) visualises the dramatic fragmentation and multi-cluster dislocation of GTPase activity in ∆*tmk3*-CRIB tip apices. (**D**) Plotting the maximum CRIB intensity and tip growth velocity against time shows that the usually rapid fluctuation of GTPase activity (t = 6.5 min) (see Figure 5D) was significantly dampened in both tips.

## Data Availability

The data presented in this study are available on request from the corresponding author. The data are not publicly available due to the considerable number and large file size of the imaging data sets as well as the requirement for specific image analyses software.

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
