# Peer review of "Stress-Activated Protein Kinase Signalling Regulates Mycoparasitic Hyphal-Hyphal Interactions in Trichoderma atroviride"

_jof, 2021, doi:10.3390/jof7050365_

Round 1
Author Response
We thank reviewer 1 for the positive evalulation. As there is no text in the "Comments and Suggestions for Authors
" section below, we assume there is no need to take any further action towards revising the manuscript.
Reviewer 2 Report
Dear authors,
Mycoparasites of the genus Trichoderma are an extremely interesting research material due to their biotechnological and commercial potential. The manuscript titled "Stress-activated protein kinase signalling regulates mycoparasitic hyphal-hyphal interactions in Trichoderma atroviride." is an interesting approach to identifying mechanisms that manage the interactions of the parasite and its host. The publication seems to be substantively coherent, the conclusions seem correct and the overall results are interesting. However, I have a few comments.
Due to the fact that most of the conclusions made are based on microscopic observations, it seems appropriate to include a chapter containing information on the basic statistical analyzes used, the number of independent experiments and the number of observations made, standard deviations, and how the statistical significance was determined. Similar data should be included in the figure descriptions (where applicable). Some figures in supplementary materials do not indicate standard deviations or the number of trials performed (n) They should be included either in the figure or in the descriptions.
The resolution of the figures should be improved. The resolution of figures and charts, in particular in supplementary materials, is insufficient. Figures should be at least 300 dpi resolution and the text and figures should not be blurry.
In Figure 6B; 8C and D; authors should consider using larger fonts in the axis descriptions and chart legend, or presenting these plots as independent figures to improve readability.
Best regards
Author Response
We thank the reviewer for encouraging us to improve our statistical analysis and data presentation. Here are our responses:
Reviewer comment #1:
Due to the fact that most of the conclusions made are based on microscopic observations, it seems appropriate to include a chapter containing information on the basic statistical analyzes used, the number of independent experiments and the number of observations made, standard deviations, and how the statistical significance was determined.
Response #1:
We added two new sections to the manuscript to explain in more detail the workflow from image data recording to quantitative analysis:
section 2.7 Live-cell imaging experimentation (now in lines 182-193)
section 2.10 Manual image data quantification and statistical analyses (now inlines 223-233).
Reviewer comment #2:
Similar data should be included in the figure descriptions (where applicable). Some figures in supplementary materials do not indicate standard deviations or the number of trials performed (n) They should be included either in the figure or in the descriptions.
Response #2:
We added the requested information to the graphs and figure text, accordingly. We also explained more thoroughly in section 2.10 why in some cases error bars were deliberately not added. This is e.g. the case in Figure S5. Please see the figure text for further details.
Reviewer comment #3:
The resolution of the figures should be improved. The resolution of figures and charts, in particular in supplementary materials, is insufficient. Figures should be at least 300 dpi resolution and the text and figures should not be blurry.
Response #3:
Thanks for flagging this up. We identified automated image compression in the programmes used to assemble the figures and manuscript file, i.e. Adobe Illustrator and MS Word, as source of the reduced quality. We now took care to deactivate compression and assembled the manuscript and supplementary data with better quality figures.
We furthermore double checked sufficient and correct resolution in the original .tiffs for each figure file and upload them to the journal. This will guarantee that the best possible image resolution will be available for the final article.
Reviewer comment #4:
In Figure 6B; 8C and D; authors should consider using larger fonts in the axis descriptions and chart legend, or presenting these plots as independent figures to improve readability.
Response #4: Font and legend sizes were increase accordingly.
We highlighted all changes in green in the revised manuscript and supplementary Word files attached to this submission.
Thanks.
This manuscript is a resubmission of an earlier submission. The following is a list of the peer review reports and author responses from that submission.
Round 1
Reviewer 1 Report
In this manuscript, the authors implement new tools to better characterize the initial interactions that underlie parasitism of fungal hyphae by the mycoparasite Trichoderma atroviride. Use of a CRIB-GFP fusion allows for the assessment of hyphae tip dynamics during the sensing, contact, and invasive stages of interaction with different hosts. Further use of this fusion in the MAP kinase mutants Tmk1 and Tmk3 provides novel insight into the roles of these signalling functions in controlling the different stages of the interaction. Key observation reported in the manuscript include the description of the distinct responses and behaviours of T. atroviride hyphae as they engage with prey hyphae (wavy growth and multi-polar growth), as well as the sorting of prey into different categories based on outcome (easy vs. impossible). More importantly, the authors were able to define specific roles for Tmk1 and Tmk3 during the interaction, with the former playing an important early contact role in signalling, whereas the latter is needed to maintain hyphae polarity during pre-contact interactions.
In my view, this manuscript provides new and significant insight into the morphological mechanisms that underlie contact between mycoparasitic fungi and their prospective hosts. I have no significant reservations about the experimental approaches employed by the authors or their interpretation of the results. I do have the following suggestions for the authors to consider;
line 201. It is unclear to me how the similarity of the recruitment pattern to that observed in N. crass supports full functionality of the construct. Loss of signal in response to induced de-polarization (e.g., via an actin depolymerizing agent) might support their point.
lines 310-318 and 388-392. If possible, these responses should be quantified.
Author Response
We thank the reviewer for the positive evaluation and constructive feedback. Please find our replies below in green.
line 201. It is unclear to me how the similarity of the recruitment pattern to that observed in N. crassa supports full functionality of the construct. Loss of signal in response to induced de-polarization (e.g., via an actin depolymerizing agent) might support their point.
Reply: Our argument is based on the identical recruitment pattern of the CRIB reporter as apical crescent/cap and part of the Spitzenkörper, in combination with its dynamic rearrangement at these localisations during directional changes of tip growth as well as its dispersal upon tip growth arrest as shown in Figure 4. Nevertheless, the suggestion to confirm CRIB function through pharamcological inhibition is a valid one and was used in our previous publication (Lichius et al., 2014). Therefore we added Supplementary Figure SX, showing that GTPase dispersal can be enforced by treatment with the Rac1-specific inhibitor NSC23766 and the F-actin polymerasion blocker Latrunculin A.
lines 310-318 and 388-392. If possible, these responses should be quantified.
Reply: Well fully agree with the reviewer that quantification of the observed phenomena is desirable. Unfortunately, we have not yet identified a suitable approach to do this in a statistically relevant manner. Please also see our reply to the similar question from reviewer 1. We would very much welcome suggestions from either reviewer how to extract reliable quantification from our imaging data, despite the general "hpyhal chaos" and difficulty to assign hyphal individuality.
Reviewer 2 Report
The authors generated a CRIB reporter and used it to examine changes in Trichoderma atroviride when it encountered with Botrytis cinerea, Rhizoctonia, and Fusarium oxysporum. They also examined the effect of TMK1 or TMK3 deletion on the CRIP reporter. Overall, it is well-presented manuscript showing different roles of TMK1 and TMK3 in the mycoparasitic interactions of T. atroviride hyphae with other fungi. (Previous studies have characterized the functions of these two MAP kinase pathways. This study is more specific for their roles in polarized tip growth)
My only major concern is related to Figure 9A. The descriptions on lines 434-438 are not clear. How often does this hyphal lysis occur? It is in the pre-contact phase. How far away from hyphae of B. cinerea or F. oxysporum? How long (incubation or confrontation) it takes to observe this phenomenon as presented in Fig. 9A? More importantly, Fig. 9A shows that all the hyphal compartments, not just the tips, had underwent lysis. To this reviewer, it is related to the function of TMK3 in stress responses (not just localized responses at hyphae tips). In this case, it is a biotic stress for T. atroviride and the opponent likely has the upper hand over the tmk3 mutant. How can the authors know there is no contact? What will happen if a dialysis membrane is placed between two fungi blocking hyphal contact but not small chemicals? (If the authors could not address these questions with more precise descriptions, it may be better to it out)
Also, is it possible that some of the observations with the tmk1 and tmk3 mutants related to their roles in defense responses against other fungi or self-/non-self recognition and signaling? Antagonistic interactions occur between many fungi that are not considered as mycoparasitic fungi. T. atroviride could be just a more aggressive fungus but it still has to defend against other fungi. Some of the results can be interpreted as defects in defense responses to other fungi. (Not necessary ‘increased sensitivities to prey defense responses’. Could be defective in defense responses) The wild-type T. atroviride may be the winner of confrontations with other fungi. The mutants may be the loser (Other fungi are not simply as offense-less ‘prey’ described in the manuscript)
The title is too vague and could be more informative . As it is, it can be mis-interpreted by readers not familiar with these MAPKs as a single signaling pathway, somehow CO but determineS. (Co-determines). Also, what is considered as onset or successful onset? Can the phrase of ‘successful onset of’ be deleted or replaced with ‘initial’?
Author Response
We thank the reviewer for the positive evaluation and constructive feedback. Please find our replies below in green.
My only major concern is related to Figure 9A. The descriptions on lines 434-438 are not clear.
How often does this hyphal lysis occur?
Reply: It is inherently difficult to provide reliable quantification of hyphal phenotypes from within the "chaotic" organisation of fungal colonies. Especially in highly branched and networked regions of the periphery and subperiphery, respectively, it is impossible to decide where one hypha ends and another one starts. Nevertheless, to provide a better idea on the frequency of the described phenotypes we added the total number of imaging experiments conducted to investigate particular phenomena together with the percentage of their observation.
It is in the pre-contact phase.
Reply: Yes. As stated in lines 431 and 435.
How far away from hyphae of B. cinerea or F. oxysporum?
Reply: The average distance beyond which hyphae of T. atroviride wild type (CRIB-GFP) show clear signs of prey-induced stress, is in the range of 1- 1.5 mm to the nearest prey hypha. For T. atroviride delta-tmk3 (CRIB-GFP) this distance is in the range of XXX mm. However, these numbers might well differ depending on the local microenvironment. Obviously, prey hyphae need to be in a physiological state to produce defense responses and diffusion must be efficient. The problem of quantification also applies in this case.
How long (incubation or confrontation) it takes to observe this phenomenon as presented in Fig. 9A?
Reply: All observations were made in a time window between 36 - 44 hours post inoculation (hpi) of the fungal confrontation samples. Live-cell imaging was always started 36 hpi (see Materials and Methods section 2.2) and typically conducted for 6-8 hours. The example in Figure 9A has been recorded exactly xx hpi, as known from the image meta data. However, the continuous development of both fungal colonies generates these phenotypes over a long period of time, depending on the available growth space and mode of interaction. Imaging along the colony edges generally presents a development and interaction gradient over time.
More importantly, Fig. 9A shows that all the hyphal compartments, not just the tips, had underwent lysis. To this reviewer, it is related to the function of TMK3 in stress responses (not just localized responses at hyphae tips).
Reply: This is precisely correct. The hyphal tip is the youngest and most sensitive part of the hypha which responds to external stresses first. The pronounced apical localisation of the CRIB reporter indicates disturbances of hyphal tip growth and changes in physiological status of individual hyphae immediately. In case the impairment is severe and the hypha cannot adapt and resume polarised tip growth, then of course, the rest of the hypha will be affected to. The reviewer's interpretation towards the lack of Tmk3 function responsible for the observed phenotype is in accordance to our interpretation.
In this case, it is a biotic stress for T. atroviride and the opponent likely has the upper hand over the tmk3 mutant.
Reply: Correct. This exactly represents our conclusion.
How can the authors know there is no contact?
Reply: We assume the reviewer refers to physical contact between hyphae of delta-tmk3 and hyphae of the prey fungus? The answer is, that we first take large overview images of the confrontation zones by image tiling of e.g. 10x4 fields of view. From these overview images we select regions of interest for detailed imaging and time course recording, and they show us what happens in the vicinity of our regions of interest. Hence, for this example we know that there was no physical contact to prey hyphae at the time of image recording. Unfortunately these large overview images (average size is about 5 GB) cannot be presented in a print medium or even in a suitable format for screen viewing. How contact between mycoparasite and prey in a more distant region might influence by means of signaling through the hyphal network is another important question that remains to be investigated.
What will happen if a dialysis membrane is placed between two fungi blocking hyphal contact but not small chemicals? (If the authors could not address these questions with more precise descriptions, it may be better to it out)
Reply: We know from our previous study using molecular size separation (Moreno et al., 2020), that prey-derived chemotropic compounds are in a range of < 3 to up to 100 kDa. We are currently testing suitable experimental set ups to address this and similar questions by live-cell imaging. Nevertheless, we are confident that the presented data provides sufficiently sound and novel insights to be published.
Also, is it possible that some of the observations with the tmk1 and tmk3 mutants related to their roles in defense responses against other fungi or self-/non-self recognition and signaling? Antagonistic interactions occur between many fungi that are not considered as mycoparasitic fungi. T. atroviride could be just a more aggressive fungus but it still has to defend against other fungi. Some of the results can be interpreted as defects in defense responses to other fungi. (Not necessary ‘increased sensitivities to prey defense responses’. Could be defective in defense responses) The wild-type T. atroviride may be the winner of confrontations with other fungi. The mutants may be the loser (Other fungi are not simply as offense-less ‘prey’ described in the manuscript)
Reply: This statement is in line with our data interpretation. We therefore see no reason to take specific action in amending the manuscript.
The title is too vague and could be more informative . As it is, it can be mis-interpreted by readers not familiar with these MAPKs as a single signaling pathway, somehow CO but determineS. (Co-determines). Also, what is considered as onset or successful onset? Can the phrase of ‘successful onset of’ be deleted or replaced with ‘initial’?
Reply: We thank the reviewer for this suggestion and changed the title to:”Stress-activated protein kinase signalling regulates mycoparasitic hyphal-hyphal interactions in Trichoderma atroviride.”